# Pharmacists’ Perceived Barriers to Human Papillomavirus (HPV) Vaccination: A Systematic Literature Review

**DOI:** 10.3390/vaccines9111360

**Published:** 2021-11-19

**Authors:** Oluwafemifola Oyedeji, Jill M. Maples, Samantha Gregory, Shauntá M. Chamberlin, Justin D. Gatwood, Alexandria Q. Wilson, Nikki B. Zite, Larry C. Kilgore

**Affiliations:** 1Department of Public Health, The University of Tennessee, Knoxville, TN 37996, USA; oonaade@vols.utk.edu; 2Department of Obstetrics and Gynecology, Graduate School of Medicine, The University of Tennessee, Knoxville, TN 37920, USA; SGregory1@utmck.edu (S.G.); NZite@utmck.edu (N.B.Z.); LKilgore@utmck.edu (L.C.K.); 3Department of Family Medicine, Graduate School of Medicine, The University of Tennessee, Knoxville, TN 37920, USA; SChamberlin@utmck.edu; 4Department of Clinical Pharmacy and Translational Science, College of Pharmacy, University of Tennessee Health Science Center, Nashville, TN 37211, USA; jgatwood@uthsc.edu; 5Preston Medical Library, Graduate School of Medicine, The University of Tennessee, Knoxville, TN 37920, USA; AQWilson@utmck.edu

**Keywords:** Human Papillomavirus, barriers, pharmacists, vaccination

## Abstract

About 45:000 cancers are linked to HPV each year in the United States alone. The HPV vaccine prevents cancer and is highly effective, yet vaccination coverage remains low. Pharmacies can play a meaningful role in increasing HPV vaccination access due to their availability and convenience. However, little is known about pharmacists’ perceived barriers to HPV vaccination. The objective of this systematic review was to summarize existing literature on perceived barriers to administering HPV vaccination reported by pharmacists. Barriers identified from selected studies were synthesized and further grouped into patient, parental, (pharmacist’s) personal, and system/organization barrier groups. Six studies were included in this review. The cost of the HPV vaccine, insurance coverage and reimbursement were commonly reported perceived barriers. Adolescent HPV vaccination barriers related to parental concerns, beliefs, and inadequate knowledge about the HPV vaccine. Perceived (pharmacist’s) personal barriers were related to lack of information and knowledge about HPV vaccine and recommendations. At the system/organization level, barriers reported included lack of time/staff/space; difficulty in series completion; tracking and recall of patient; perceived competition with providers; and other responsibilities/vaccines taking precedence. Future strategies involving pharmacy settings in HPV-related cancer prevention efforts should consider research on multilevel pharmacy-driven interventions addressing barriers.

## 1. Introduction

Human Papillomavirus (HPV) is estimated to be the cause of 70% oropharynx, vaginal, and vulvar cancers, 60% of penile cancers, and 90% of anal and cervical cancers [1]. HPV-associated cancers are the only known cancers that can be prevented by receiving a vaccine. Yet, there are over 45,000 newly diagnosed HPV-related cancers in the United States each year [1]. The HPV vaccine has been shown to be safe and highly effective [2]. The Centers for Disease Control and Prevention (CDC) established HPV vaccination as a public health priority, yet vaccination coverage rates are less than the Healthy People 2020 goal that 80% of adolescents complete the vaccine series [3,4]. Nationally in 2017, less than 66% of adolescents received the first dose of the HPV vaccine, and only about 49% completed the series [5]. The HPV vaccine dosing schedule consists of a series of two or three doses, depending on the age of the patient at the start of the schedule [6]. For example, a patient that is under the age of 15 is recommended to receive two doses of the HPV vaccine administered 6–12 months apart. A patient starting the vaccine series on or after the 15th birthday is recommended to receive three doses of the HPV vaccine [6]. In this dosing schedule, the second vaccine should be administered 1–2 months after the first dose, and the third dose should be administered 6 months after the first dose [6].

Community pharmacies have the potential to play a meaningful role in increasing HPV vaccination rates. The President’s Cancer Panel [7] and National Vaccine Advisory Committee [8] released statements urging the increase in uptake of HPV vaccination rates by using pharmacies as a strategic site. Pharmacies are conveniently located for most families, including those in rural communities. In fact, most residents in the United States (91%) live within five miles of a pharmacy [9]. Additionally, pharmacies are typically open for extended hours and on weekends. Given the expanding scope of practice and evolving role of the pharmacist and pharmacy as playing a vital role in public health, due to their accessibility, cost-efficacy, and ability to provide education and shared responsibility, along with patient acceptance, they have been identified as crucial partners in vaccination administration [10].

During the ongoing COVID-19 pandemic, the U.S. Department of Health and Human Services (HHS) declared that pharmacists are allowed to administer childhood vaccines for children that are three years and above [11]. Most state laws also allow pharmacists to administer HPV vaccine to patients, with some states having varying age restrictions and/or prescription requirements (only two states do not allow pharmacists to administer the HPV vaccine at all) [12]. Pharmacists provide a convenient option for patients to receive the HPV vaccine. However, little is known about the current perceptions and barriers to administrating HPV vaccination among pharmacists. Therefore, this review aims to systematically examine the literature on pharmacists’ perceived barriers to administering the HPV vaccine as a mechanism for improving HPV vaccination uptake.

## 2. Methodology

This systematic review was conducted following the Preferred Reporting Items for Systematic Reviews and Meta-Analyses statement (PRISMA) [13].

### 2.1. Search Strategy

Electronic searches were created and completed by a research team member and health sciences librarian (A.W.) experienced with systematic searches in the following databases: PubMed [NLM], Web of Science [Thomson Reuters], Cochrane Central Register of Controlled Trials [Wiley], and Scopus [Elsevier]. The search strategy was created first in PubMed (Appendix A), and then translated for each database platform as applicable. MeSH terms and keywords were used to search concepts related to Human Papillomavirus, vaccines, barriers, and pharmacists. The searches were performed on 18 September 2020. Results were restricted to English language. There were no restrictions set on the year of publication for inclusion. Duplicate references were removed using Rayyan software [14]. The references cited in the included articles were reviewed for additional relevant articles.

### 2.2. Study Selection & Data Extraction

The titles and abstracts were first screened independently for eligibility by two researchers (S.G., O.O.). Then, the same researchers reviewed the full text of selected abstracts for further eligibility. Conflicts were resolved by a third researcher (J.M.M.). Study inclusion criteria include: (1) primary research article including qualitative, quantitative, and mixed-method studies, (2) reported outcome included perceived barrier to administering HPV vaccine (3) study population included pharmacists, pharmacy students, pharmacy technicians, and pharmacist representatives. Studies were excluded if the study objectives were unrelated to HPV vaccination. Two co-authors (O.O., J.M.M.) abstracted and synthesized all findings. Barrier type categorization was adapted from the findings reported in the included studies [15,16,17]. Some barriers extended, at least partially, across other barrier types. All barrier type classifications were discussed among co-authors, and a consensus was made for the final classification. Another co-author (S.M.C) reviewed the findings independently for accuracy. Any conflicts were resolved by deliberations among the co-authors.

## 3. Results

### 3.1. Literature Search and Study Characteristics

The electronic database searches yielded 444 articles; 292 titles and abstracts were selected, and 13 full-text studies were reviewed for inclusion. In the end, 6 out of 13 studies were selected for this review using the exclusion and inclusion criteria [15,16,17,18,19,20]. Figure 1 illustrates the PRISMA diagram detailing the search and selection process. Four of the studies [15,16,18,20] examined barriers related to HPV vaccine alone, while the remaining two studies [17,19] included other vaccines in addition to HPV. All included studies were carried out in the United States; however, geographical location and pharmacy setting differ across studies (Table 1). Table 2 presents an overview of the study designs, methodologies, and data reporting. While all the studies utilized a cross-sectional study design, the methodologies and results reporting varied widely across studies (Table 2). Table 3 presents a summary of the key findings for the included studies. These findings were further grouped by level of barrier into patient, parental, (pharmacist’s) personal, and system/organization barrier groups, which were adapted from the findings reported in the included studies (Table 4) [15,16,17]. Some barriers extended, at least partially, across other barrier types. For example, “financially-related” (or cost) barriers were reported across multiple barrier types [15,17,18,19,20]. To the patient a “financially-related” (or cost) barrier may be the out-of-pocket cost of the vaccine, which is distinct from the “financially-related” (or cost) barrier at the systems/organizational level, which may be more related to the financial burden of stocking the vaccine. 

### 3.2. Barrier Levels

Patient: Four out of the six included studies reported patient-related barriers [15,16,17,19]. The most commonly reported barriers at the patient level were inadequate demand [15,16] and patient refusal [17]. Reasons for patient refusal reported in one of the studies included perceived safety concerns and financial reasons [17]. Other barriers included patients lacking insurance coverage [17] and difficulty overcoming financial barriers for an adolescent [19].

Parental: Three studies included parent-related barriers [15,18,20]. Across these studies, certain parental concerns and perceptions were reported as pharmacists perceived barriers to HPV vaccination. This included safety and efficacy concerns about the HPV vaccine [15], concerns that agreeing to vaccination means they are condoning premarital sex [15], and concerns that their children will engage in riskier sexual behaviors if they receive the vaccine [15]. Parental beliefs that their children are not at risk for HPV infection and that children are not old enough for the HPV vaccine were also reported as barriers [15]. Other key parent-related barriers reported include inadequate demand from parents [20], parental believe the cost is too high [15], perceived stigma about vaccination among parents of adolescents [18], parental consent [18], lack of knowledge [20], and inadequate education/understanding about HPV infection [15].

(Pharmacist’s) Personal: Three studies reported pharmacist’s personal barriers to HPV vaccination. [16,18,19]. In one of the studies, lack of information was reported as a barrier [16]. This included inadequate knowledge to educate and recommend the HPV vaccine and also a lack of information on subjects such as cost and storage of the vaccine [16]. Misinformation among pharmacists regarding HPV vaccination and administration was also reported as a barrier [16]. For instance, one pharmacist believed (incorrectly) that the HPV vaccination coverage in their (rural) area was good [16]. Another example of misinformation was evidenced by a pharmacist stating that Medicaid does not allow pharmacists to provide the HPV vaccine to those under 18, even with a prescription [16]. A third example of misinformation was evidenced by a pharmacist stating that adolescents are supposed to receive the HPV vaccine only at their doctor’s office [16]. The perception that HPV infection/mode of transmission is a sensitive subject was also cited as a barrier [16]. Other personal barriers include safety concerns about the potential adverse reaction after receiving the HPV vaccine [16,18], and the belief that the Advisory Committee on Immunization Practices (ACIP) recommendations are controversial [19].

System/Organization: Five out of the six studies reviewed reported system/organization level barriers [15,16,17,18,20]. Three of these studies reported lack of time and staffing as barriers to HPV vaccine administration [16,17,20]. For example, often times only one pharmacist may be present, which could result in major interruptions in work flow to stop and administer an HPV vaccine [16]. Two studies reported that a lack of physical space was also a barrier [16,20]. For example, a lack of consulting space to accommodate both parent and adolescent for HPV vaccine counseling and/or administration is a barrier [16]. Financial barriers related to vaccination cost, insurance reimbursement, and compensation were also frequently cited [15,17,18,20]; this included lack of insurance coverage of vaccine costs [15], difficulty in knowing whether insurance will reimburse for vaccine [17], and adequate compensation for vaccine purchase/supplies purchase/vaccine administration [17]. Other cost-related barriers included upfront cost of buying vaccine/supplies and lack of demand to justify cost of stocking vaccines [17].

At the organization level, other responsibilities/vaccines taking precedence over HPV vaccination was also reported as a barrier [17,20]. Other reported barriers were related to series completion which included lack of incentives for series completion [15,17], tracking and recall of patients [18], and remembering to screen patients for vaccine [17].

In one study, concerns that it may appear that pharmacists are in competition with health care providers to administer the vaccine was seen as a barrier [16]. Liability issues relating to adverse effects after vaccination [16], lack of educational materials to provide to parents [20]; and inadequate vaccine promotion/education were also reported as barriers [18].

## 4. Discussion

Community pharmacies could potentially play an important role in improving HPV vaccination rates due to their availability and accessibility. This review presents a summary of the literature on pharmacists’ perceived barriers to providing the HPV vaccine in a pharmacy setting. Based on the findings from the six included studies, pharmacists’ perceived barriers to HPV vaccination uptake exist at the patient, parental, pharmacist’s (personal), and system/organization levels. The existence of reported barriers at different levels suggests that efforts at improving HPV vaccination rates among pharmacies should be targeted at these levels.

Barriers related to the cost of the HPV vaccine and insurance coverage were frequently cited in the articles reviewed for this paper [15,17,18,19,20]. This is in concert with previous studies where vaccine cost and lack of insurance coverage have also been reported as specific barriers to HPV vaccination among health care providers [23,24]. Other studies have also reported that financial reasons may be a parental barrier for HPV vaccination for their children [25]. In fact, one study found that a lower proportion of parents were willing to vaccinate their children if the vaccine was not covered by insurance [25]. A number of options are available to cover the cost of HPV vaccine. Under the Affordable Care Act (ACA), most private insurance plans in the United States are required to cover immunizations recommended by ACIP without consumer cost-sharing [7]. Additionally, the Vaccine for Children (VFC) program is a federally funded program in the United States that covers vaccine costs for eligible children below 19 years of age (eligibility criteria include Medicaid-eligible, uninsured, underinsured, and American Indian or Alaska Native) [26]. A study among physicians reported that those participating in VFC program were less likely to indicate cost as a barrier to HPV vaccination [24,27]. Previous research showed that the VFC program led to increased vaccination for certain vaccines [28]. However, barriers such as state laws (only 34 states allow pharmacies to participate in VFC), inadequate reimbursement to cover actual costs, administrative burden and low demand from eligible persons may affect pharmacies’ participation in VFC or willingness to carry all vaccines [7,29]. Other sources that cover HPV vaccine cost in the United States include Merck assistance programs [30], Medicaid and Children’s Health Insurance Program (CHIP) [31]. Promotion and public education on these little or no-cost options for HPV vaccination could improve access.

Parents play an important role in HPV vaccination for their adolescent children. Results from this review showed that pharmacists perceived parental concerns, beliefs, and lack of knowledge as barriers to HPV vaccination [15,18,20]. Similar findings were reported from a study that examined perceived barriers to HPV vaccination among health care providers; the authors reported parental beliefs and misconceptions as a major barrier to HPV vaccination [32]. About one-third of parents in previous research reported that they were willing to allow their children to get the HPV vaccine in a pharmacy setting [33]. Likewise, another study showed that 44% of parents are willing to get an HPV vaccine from pharmacies for their children [34]. Improving parental awareness and clarity that pharmacists offer HPV vaccines for children may facilitate series completion [33]. Also, strategies to educate parents on the importance of the HPV vaccine and demystify misconceptions may positively influence parental decision to vaccinate their children. This may include additional training for pharmacists on ways to educate and provide effective HPV vaccine recommendations to parents [15].

Partnerships and collaboration agreements between health care providers and pharmacists, including strong provider recommendation and referral to pharmacies for subsequent HPV vaccine doses may increase parental awareness and vaccine uptake [33]. Physicians could help improve parental awareness as parents may prefer to learn about the availability of pharmacists to provide the HPV vaccination from their children’s physicians [33]. However, there is a lack of uniformity across states in the US and diverging opinions regarding authority to administer vaccines to children and adolescents that complicates the theoretical partnerships and collaboration agreements between health care providers and pharmacists. For instance, laws and regulations granting pharmacists the authority to administer HPV vaccine to all or certain adolescent age groups vary across states [12]. Also, there is a lack of consensus among professional organizations, like the American Academy of Pediatrics (AAP), and the US Department of Health and Human Services regarding whether pharmacists should be authorized to administer vaccines to children and adolescents [11,35]. In 2020, AAP released a statement that their organization believes that children should receive vaccines from a pediatrician, which opposes HHS authorization that allows pharmacists to administer childhood vaccines [36].

Three studies included in this review reported personal barriers among pharmacists to HPV vaccine administration [16,18,19]. The personal barriers were not religious or moral, but related primarily to inadequate knowledge to provide recommendations for HPV vaccination, or educate patients about the HPV vaccine (considering the sensitivity of HPV-related subject) [16,19]. This is promising as additional training on HPV vaccine recommendation strategies, coverage, and administration for practicing pharmacists, as well as student pharmacists, could help improve the pharmacists’ knowledge and comfort level in educating and administering the HPV vaccine [37].

At the organization level, remembering to screen [17] as well as tracking and recall of patients were identified as barriers to HPV vaccination [18]. In one of the studies reviewed, only 33 percent of participants reported that they utilized their state’s Immunization Information Systems (IIS) [18]. IIS can ensure timely immunization, clinical decision support, records consolidation, and data exchange among health care providers [38]. However, the operating and reporting for IIS requirements vary by state [39]. Another key organization level barrier reported was other responsibilities/vaccines taking precedence [17,20] over HPV vaccination and lack of time/staff/space [16,17,20]. The decision to prioritize and offer HPV vaccines by pharmacies may depend on the goals of the organization/management and may differ between independent versus chain pharmacies.

While the purpose of this review was to summarize existing literature related to the perceived barriers to administering HPV vaccination reported by pharmacists, some of the studies reported facilitators of HPV vaccination suggested by study participants [16,18,20]. Most frequently suggested facilitator was education about HPV vaccine [16,18,20]. As reported in one of the studies, education may involve providing education to patient/parent, pharmacists, and health care providers [20]. Another suggested way to improve HPV vaccination is advertising specifically through mass/social media [16,20]. Pharmacist providing stronger recommendation to patients about getting HPV vaccination was also reported as potential facilitator of HPV vaccination [20]. Partnership involving health care providers, schools [16,20] and public health organization [16] was reported as a facilitator. For instance, doctors may recommend that their patient get the HPV vaccine in a pharmacy or schools holding vaccine clinics [20]. Other facilitators reported by pharmacists include ease of accessibility of community pharmacies [16], clear guidelines from pharmacy/corporate management [18], vaccine promotion within and outside the pharmacy [18], adequate insurance coverage for vaccination [20], and state legislative authority to provide vaccination [18].

Evidence-based strategies to improve HPV vaccination rates that have previously been implemented in other settings, such as in primary care, may help to improve HPV vaccination in pharmacies. For instance, previous studies suggest that reminder and recall systems [40] and decision support systems [41] may be effective in improving HPV vaccination, which potentially could facilitate series completion. Alternatively, successful evidence-based strategies implemented in pharmacy settings to improve other vaccines could also be adapted to improving HPV vaccination rates. In one study, pharmacy-based interventions, including newspaper press releases, use of flyer advertisement, and personalized letters about herpes zoster infection and vaccination were shown to improve herpes zoster vaccination rates [42]. In another study, individuals who received a phone call intervention from pharmacists were more likely to receive their second dose of the recombinant zoster vaccine, thereby facilitating series completion [43]. Further research is needed to explore interventions unique and specific to improving HPV vaccination and series completion in pharmacies.

To our knowledge, this is the first study to review previously reported perceptions of barriers to HPV vaccine administration among pharmacists. Numerous commentaries and professional organizations suggest that pharmacies could play an important role in improving HPV vaccine uptake. The studies included in this review varied greatly in terms of study design, which made the synthesis of findings challenging. The search strategy employed to identify eligible studies for this review, encompassed publications on studies conducted worldwide. However, all the eligible studies included in this review were conducted in the United States. The perceived barriers that pharmacists experience may vary geographically within and outside of the United States. Therefore, generalizability of results may be limited. Nevertheless, many of these barriers are, to some extent, universal and therefore may provide direction for those investigating HPV vaccination barriers in different health care delivery systems. This literature search yielded only a few studies, which indicates that additional research is needed to explore pharmacist’s perceived barriers, especially at the system/organization levels. Because most of the studies included in this review relied primarily on quantitative survey-based data collection methods, it limits the ability to thoroughly understand pharmacists’ perceived barriers to HPV vaccine administration. A more comprehensive understanding of barriers to HPV vaccine administration is needed, especially to better understand how these barriers may impact various pharmacies in utilizing differing immunization delivery practices. Future work should include in-depth qualitative analyses of barriers among pharmacists practicing in a wide variety of settings.

In conclusion, pharmacies present an opportunity to increase HPV vaccination rates. Targeted public health efforts to increase HPV vaccination among pharmacies should consider a multilevel approach. Future strategies involving pharmacy settings in HPV-related cancer prevention efforts should consider research on pharmacy-driven interventions that addresses the barriers to administering the HPV vaccine at various levels.

## Figures and Tables

**Figure 1 vaccines-09-01360-f001:**
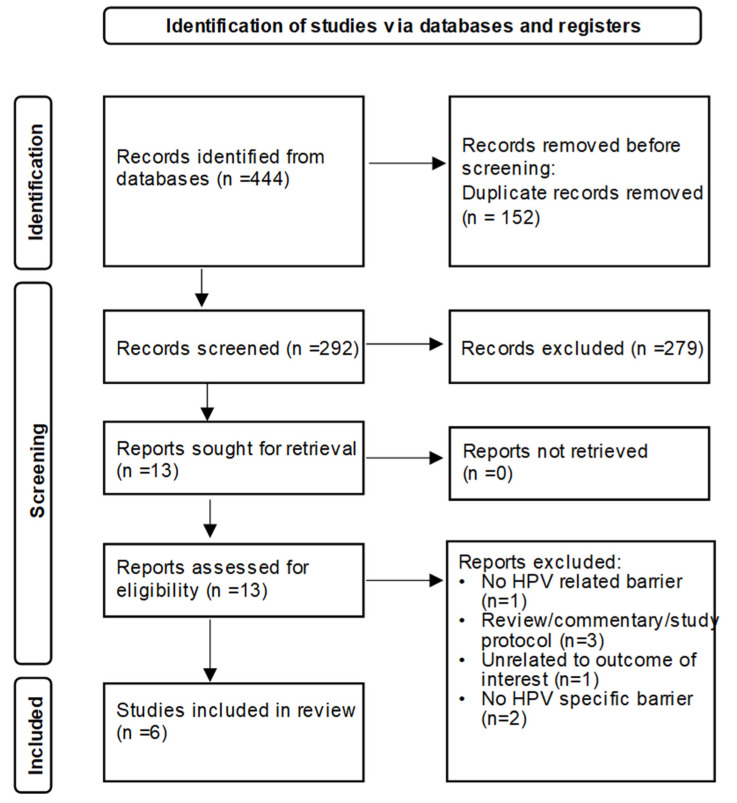
PRISMA diagram showing search and selection.

**Table 1 vaccines-09-01360-t001:** Characteristics of included studies.

Author, Publication Year	Sample Size	Study Population Characteristics	Pharmacy Setting	Vaccines Examined
Berce et al., 2020 [17]	236	Pharmacists located in Wisconsin (USA). 79% worked in pharmacies that were primarily located in urban counties (79%), 7% were in rural counties and 14% were located in multiple counties and/or in both rural and urban areas	Health system community (29%), chain community (27%), independent (26%), ambulatory care clinic (11%), inpatient (7%)	HPV, Influenza, Zoster, Pneumococcal, Tdap/Td, DTap, Hepatitis A and B, Hepatitis A, Meningococcal, MMR, Varicella, Polio and others
Hastings et al., 2017 [15]	154	One participant represented each pharmacy which included pharmacy owners, managers, or staff pharmacists located in Alabama (USA)	Chain pharmacy (53%), and independently owned (47%)	HPV
Islam et al., 2019 [18]	40	Pharmacists from 8 states (Alabama, Indiana, California, Maine, Kentucky, Tennessee, Texas, and Washington) in the USA that previously or currently were administering HPV, Meningococcal, Tdap or TD vaccines to adolescents	Chain (78%), independent (13%), grocery (5%), big box retailer (5%)	HPV
Ryan et al., 2020 [16]	11	Pharmacists in 7 rural counties in Iowa (USA)	Independently owned (100%)	HPV
Skiles et al., 2011 [19]	24	Pharmacy association directors or designees from all 50 states in the USA were targeted (92% were pharmacists)	Information not reported	HPV, Tdap, Influenza
Tolentino et al., 2018 [20]	240	Community pharmacists in Utah (USA)	Community/outpatient (80%), ambulatory care clinic (10%), community outpatient and inpatient (8%), other (3%)	HPV

Td-Tetanus/Diphtheria, Tdap-Tetanus/Diphtheria/Pertussis, MMR- Measles/Mumps/Rubella, Other- Typhoid, Yellow fever, Japanese Encephalitis and not otherwise stated.

**Table 2 vaccines-09-01360-t002:** Overview of included study methodologies and data reporting.

Author, Publication Year	Study Design	Data Collection Tool	Data Reporting
Berce et al., 2020 [17]	Cross-sectional survey design using an anonymous electronic survey	Modified version of a previous national physician survey [21] on barriers to adult vaccination which asks respondents to classify multiple potential barriers to immunization on a 4-point response scale.	Barriers were grouped by pharmacies that provide and do not provide immunization.
Hastings et al., 2017 [15]	Cross-sectional survey design using a modified version of Dillman’s Tailored Design Method of survey administration [22]	A 65-item survey that took approximately 15 min to complete. Measures were categorized into 5 sections: 1. key informant and pharmacy site demographic characteristics; 2. general vaccination services and strategies used to increase HPV vaccine uptake; 3. pharmacists’ perceptions of HPV and the vaccine; 4. perceived system barriers to the provision of HPV vaccinations; and 5. perceived parental reasons for HPV vaccine hesitancy. Most of the questions were 5-point Likert-type rating scales. Questions measuring HPV and the vaccine perceptions were adapted from an existing instrument. Questions assessing system barriers were informed by previous research.	Descriptive statistics were used to describe participants characteristics, vaccine practices, barriers and attitudes.
Islam et al., 2019 [18]	Cross-sectional study design using semi-structed interview to complete a survey	Survey items included 52 close-ended questions and 24 open-ended questions to examine pharmacists insights into administering vaccines. Interviews lasted 30–60 min.	Semi-qualitative responses were analyzed using thematic analysis, to create response categories and then coded using descriptive frequency statistics.
Ryan et al., 2020 [16]	Cross-sectional study design using interview.	Interview guide using questions and concepts adapted from a previous project that included the following topics: the role of rural, independent pharmacists in HPV vaccine promotion and uptake; willingness to educate parents, refer patient, and administer the HPV vaccine; priority of HPV vaccine promotion, and vaccination barriers and facilitators in the pharmacy and the community	Interview responses were analyzed using thematic analysis.
Skiles et al., 2011 [19]	Cross-sectional study design using telephone interviews to complete a survey	Survey questions asked about immunization practices, vaccine beliefs, minor consent issues, and minor consent laws. Responses to the attitude and/or belief questions were measured on a 5-point Likert scale.	Attitude and belief responses were collapsed to a dichotomous response for analysis. Differences in attitudes across vaccines were tested using score test on the basis of the generalized estimating equation for the generalized linear model.
Tolentino et al., 2018 [20]	Cross-sectional study design using an anonymous electronic survey	A 73-item survey adapted from an HPV vaccination survey previously conducted with Utah primary care providers that asked about HPV vaccination knowledge, attitudes about the HPV vaccine, behavior for recommending the vaccines (HPV, influenza, meningococcal disease, Tdap), and barriers for adolescents’ vaccination. Survey questions were multiple choice, true/false, and Likert scale.	Descriptive statistics were used to analyze the demographics of pharmacists, as well as their knowledge and attitudes regarding the HPV vaccine, vaccine recommendation levels and strategies, and self-identified barriers to vaccine recommendations. Content analyses were used to identify the themes.

**Table 3 vaccines-09-01360-t003:** Key findings of included studies.

Author	Summary of Findings
Skiles et al. [19]	96% reported that financial challenge is a barrier to HPV vaccination access for adolescents (*p* < 0.001); 75% of participants reported that access to HPV vaccine is moderately to extremely difficult (*p* = 0.030); 67% reported that ACIP recommendations are moderately to extremely controversial in the community (*p* < 0.001)
Hastings et al. [15]	Participants reported the following as very/extremely likely to be a system-related barrier to HPV vaccination: lack of demand (56.5%), failure of cost coverage by insurance (54.8%), vaccine expiration before use (54.1%), difficulty ensuring patients are completing the necessary 3 doses (39.9%), and lack of adequate reimbursement (38.4%). Participants reported that they somewhat agree/strongly agree that the following are parent-related barriers to HPV vaccination: lack of education (86.6%) safety concerns (78.7%), reluctancy to talk about sexuality/sexually transmitted infections (76%), concerns that agreeing to vaccination means they support premarital sex (67.3%), concerns about efficacy of vaccine (64.6%), cost (53.3%), believe that their children are not at risk (67.3%), believe that their children are too young (65.3%), concern that children will practice riskier sexual behaviors (58.7%).
Berce et al. [17]	Insurance and time/priority were reported as largest barrier. Compared with those that do not immunize, financial barrier was larger among those that do immunize (*p* = 0.022). Barriers reported among those that do immunize included patients having insurance coverage for vaccines (90%), patients refusal due to financial reasons (89%), patients refusing vaccine (89%), determining insurance reimbursement (87%), other responsibilities taking precedence (84%), patient refusal due to perceived safety issues (79%), lack of staff (78%), remembering to screen patients (76%), having enough demand to justify the cost of stocking vaccines (71%), upfront cost of buying vaccines and supplies (55%), adequate compensation for administration (72%), and adequate compensation for product (68%) and supplies (58%) purchase. Barriers reported among those that do not immunize included other responsibilities taking precedence over vaccinating (94%), patient refusal (72%), patient refusal due to perceived safety issues (67%), determining insurance reimbursement (66%), lack of staff (61%), remembering to screen patients (60%), patient having insurance coverage (57%) and adequate compensation for administration (53%).
Ryan et al. [16]	Barriers were grouped into personal and organizational barrier. Personal barriers included sensitivity on the subject of HPV infection, lack of information, safety concerns, misinformation about HPV vaccination coverage and access. Organization barriers include lack of time and staff, liability issues relating to adverse effect after vaccination, low number of adolescents coming to the pharmacy, and competition with local health care providers.
Tolentino et al. [20]	Barriers reported included lack of parental knowledge, parental concerns/opposition, lack of educational materials for parents, high copay, lack of demand from parents, lack of time and space, high priority for other vaccines compared with HPV, and lack of incentive for series completion.
Islam et al. [18]	Major barriers to providing HPV vaccines to adolescents included the following: parental consent (28%), tracking and recall of patients (17%), stigma about vaccination among parents (17%), education/vaccination promotion (17%), cost of vaccination (11%), potential adverse reactions (11%).

**Table 4 vaccines-09-01360-t004:** Summary of key barriers grouped by level of barrier.

Author, Year	Barrier Levels			
	Patient	Parental	Personal	System/Organization
Skiles et al., 2011 [19]	Financial barriers Access to HPV vaccination is difficult		Belief that ACIP recommendations are controversial	
Hastings et al., 2017 [15]	Too few patients who want the HPV vaccine	Safety concerns about HPV vaccine Concerns that agreeing to vaccination means they are condoning premarital sex Efficacy concerns about HPV vaccine Lack of education/understanding about HPV infection Parental belief that their children are not at risk for HPV infection Reluctancy to discuss sexuality/sexually transmitted infections Parental belief that their children are too young for the vaccine Concern that their children will practice riskier sexual behaviors if they receive the vaccine Parental belief that the cost of vaccine is high		Lack of coverage of vaccination cost by some insurance companies Vaccine expiration before use Difficulty ensuring series completion Lack of adequate reimbursement
Tolentino et al., 2018 [20]		Lack of parental knowledge Parental concerns/opposition Lack of demand from parents		Lack of time and space High priority for other vaccines compared with HPV Lack of incentive for series completion High copay Lack of educational materials for parents
Islam et al., 2019 [18]		Parental consent Perceived stigma about vaccination among parents of adolescents	Potential adverse reaction	Tracking and recall of patients Cost of vaccination Education/promotion of vaccination
Berce et al., 2020 [17]	Patient refusing vaccine Patient refusal due to perceived safety issues Patients having insurance coverage for vaccine Patient refusing vaccines for financial reasons			Other responsibilities taking precedence over vaccinating Determining if patient’s insurance with reimburse Having enough staff to provide vaccine Remembering to screen patients for needed vaccine Adequate compensation for vaccine administration Adequate compensation for vaccine product purchase Adequate compensation for supplies purchase. Upfront cost of buying vaccines and supplies Having enough demand to justify the cost of stocking some or all recommended vaccines
Ryan et al., 2020 [16]	Low number of adolescents coming to the pharmacy		Sensitivity of subject Lack of information Safety concerns Misinformation	Lack of time, staff and space Liability issues Competition with local health care providers

## Data Availability

Not applicable.

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
