# Peer review of "Pharmacists’ Perceived Barriers to Human Papillomavirus (HPV) Vaccination: A Systematic Literature Review"

_vaccines, 2021, doi:10.3390/vaccines9111360_

Round 1

Reviewer 1 Report

This paper is of interest and provides a useful synthesis of the available data on pharmacists' perceived barriers to HPV vaccinations.  Addressing barriers to HPV vaccine uptake will be essential to improving HPV vaccine coverage and enhance our fight against cervical cancer - a preventable disease which is currently marred by ethnic and socioeconomic inequity.   

Strengths:  
1. The review was undertaken by a researcher experienced in systematic literature searches and following the PRISMA process.
2. While the paper included a lot of detail which is useful for those interested in the specifics, the authors summarized the key barriers well for those more interested in the take home message.

Weaknesses:
1. Obviously the review was based on Pharmacists' perceptions of barriers so it is possible that there are other barriers which are important to parents and adolescent children but which have not been identified.  However, the aims of the study are clear and pharmacists are in a unique position to identify barriers which are not necessarily obvious to other groups of interest.
2.  Some of the barriers may be quite specific to the US environment - however, this is clearly noted in the discussion and many of the barriers are somewhat universal and therefore good starting points for researchers interested in investigating barriers in their own health care environment. 

typos:

page 9, final paragraph: willing to vaccinate...

page 11, first paragraph: following "...HPV vaccination [20]" - replace comma with full stop.

Author Response

Reviewer 1

Point 1:

This paper is of interest and provides a useful synthesis of the available data on pharmacists' perceived barriers to HPV vaccinations.  Addressing barriers to HPV vaccine uptake will be essential to improving HPV vaccine coverage and enhance our fight against cervical cancer - a preventable disease which is currently marred by ethnic and socioeconomic inequity.   

Strengths:  
1. The review was undertaken by a researcher experienced in systematic literature searches and following the PRISMA process.
2. While the paper included a lot of detail which is useful for those interested in the specifics, the authors summarized the key barriers well for those more interested in the take home message.

Response 1: Thank you for taking time to review the manuscript and provide feedback.

Point 2:

Weaknesses:
1. Obviously the review was based on Pharmacists' perceptions of barriers so it is possible that there are other barriers which are important to parents and adolescent children but which have not been identified.  However, the aims of the study are clear and pharmacists are in a unique position to identify barriers which are not necessarily obvious to other groups of interest.
2.  Some of the barriers may be quite specific to the US environment - however, this is clearly noted in the discussion and many of the barriers are somewhat universal and therefore good starting points for researchers interested in investigating barriers in their own health care environment. 

Response 2: We agree with Reviewer 1 that there are several very important barriers to HPV vaccine uptake perceived by parents and adolescent children, which was beyond the scope of this review. Also, a number of publications have reported perceived barriers to HPV vaccination among other healthcare providers, but we agree with this Reviewer that the focus on pharmacist’s perceived barriers is unique. 

We agree with Reviewer 1 that some of the barriers are very specific to the US environment. The fact that these barriers are somewhat universal is a good point and we have updated lines 297-304 to reflect this, which now reads:

The search strategy employed to identify eligible studies for this review, encompassed publications on studies conducted worldwide. However, all the eligible studies included in this review were conducted in the United States. The perceived barriers that pharmacists experience may vary geographically within and outside of the United States, therefore, generalizability of results may be limited. Nevertheless, many of these barriers are, to some extent, universal and therefore may provide direction for those investigating HPV vaccination barriers in different health care delivery systems.

Point 3:

typos:

page 9, final paragraph: willing to vaccinate...

page 11, first paragraph: following "...HPV vaccination [20]" - replace comma with full stop.

Response 3: We have corrected the indicated typos in line 199 (by including the word “to”) and in line 271 (by replacing the comma with a period).  

Reviewer 2 Report

Dear Author,

I read with interest your Systematic Literature Review on Perceived Barriers to Human Papillomavirus Vaccination.

I believe that the quality of the contents, the clarity of the results presentation (especially Table 4 is really useful to synthesize the main findings), the discussion section are really well-written and interesting for Public Health and for future vaccination offer strategies also in Europe.

Regarding some limitations I could suggest to the authors to add (and consequently Justify it)
1. The large majority of the studies included are limited to the US territory (due to the prevalent diffusion of this good practice in this Area).
2. A metanalysis was not conducted probably due to the limited number obtained in the SLR (6 studies)

Author Response

Reviewer 2

Point 1:

Dear Author,

I read with interest your Systematic Literature Review on Perceived Barriers to Human Papillomavirus Vaccination.

I believe that the quality of the contents, the clarity of the results presentation (especially Table 4 is really useful to synthesize the main findings), the discussion section are really well-written and interesting for Public Health and for future vaccination offer strategies also in Europe.

Response 1: Thank you.

Point 2:

Regarding some limitations I could suggest to the authors to add (and consequently Justify it)

  1. The large majority of the studies included are limited to the US territory (due to the prevalent diffusion of this good practice in this Area).

Response 2: Yes, this is correct that the studies meeting eligibility criteria were conducted in the US, despite geographic location not being an inclusion/exclusion criterion. This indeed limits generalizability. We have added additional information in the Discussion section that describes this in more detail.  That section (lines 297-302) now reads:

The search strategy employed to identify eligible studies for this review, encompassed publications on studies conducted worldwide. However, all the eligible studies included in this review were conducted in the United States. The perceived barriers that pharmacists experience may vary geographically within and outside of the United States, therefore, generalizability of results may be limited.

Point 3:

  1. A metanalysis was not conducted probably due to the limited number obtained in the SLR (6 studies)

Response 3: We agree, a meta-analysis would not be possible given the heterogeneity of reported barriers (outcome measures) and study designs employed across the small number of studies included in the review.